# The Impact of Antibiotics Administration on Mortality for Time in Sepsis and Septic Shock Patients including Possible Reasons for Delayed Administration in Malaysia

**DOI:** 10.3390/antibiotics11091202

**Published:** 2022-09-05

**Authors:** Ann L. Arulappen, Monica Danial, Ling Wei Ng, Jui Chang Teoh

**Affiliations:** 1Pharmacy Department, Hospital Seberang Jaya, Ministry of Health, Seberang Jaya 13700, Malaysia; 2Clinical Research Center, Hospital Seberang Jaya, Ministry of Health, Seberang Jaya 13700, Malaysia; 3Department of Anesthesiology, Hospital Seberang Jaya, Ministry of Health, Seberang Jaya 13700, Malaysia

**Keywords:** antibiotic, timing, severe sepsis, septic shock, mortality

## Abstract

The 2017 Surviving Sepsis Campaign guidelines endorse a focus on the rapidity of treatment once sepsis has been identified, with a strong recommendation for the administration of antimicrobial drugs within one hour; however, the quality of the supporting evidence is evaluated as moderate. This study was conducted for six months prospectively at a single center in an intensive care unit (ICU) from March 2020 to August 2020. All the patients, regardless of their age and gender, admitted into ICU who had their first episode of sepsis or septic shock concomitantly started on a broad-spectrum antibiotic given intravenously. For patients who had multiple episodes of sepsis throughout the study period, data from the very first episode of the sepsis were included in this study. Of all the 78 patients, only 38 (48.7%) received the antibiotics prescribed within an hour. The compliance rate as per the Surviving Sepsis Campaign was only 51.3%, which accounted for 40 patients. The overall survival rate was 60.3%. This study revealed that delayed antibiotics administration (more than an hour) significantly affects mortality.

## 1. Introduction

Sepsis is a common, life-threatening organ dysfunction driven by a dysregulated host response to infection. Outcomes have improved over the years in line with a focus on intravenous fluids, appropriate antimicrobials, and other supportive measures. Nevertheless, the mortality rate remains at 30% to 50% for septic shock [1]. The 2017 Surviving Sepsis Campaign guidelines endorse a focus on the rapidity of treatment once sepsis has been identified, with a strong recommendation for the administration of antimicrobial drugs within one hour; however, the quality of the supporting evidence is evaluated as moderate [2].

Data from multiple observational studies have addressed the question of whether early antibiotic administration is associated with improved survival. It is widely accepted and biologically plausible that giving antibiotics as early as possible to patients with sepsis should improve their outcomes [3,4]. Kumar et al. and Vincent X et al. reported that early administration of antibiotics was significantly associated with lower mortality rates. On the contrary, a study conducted in Japan concluded that there is no association between earlier antibiotic administration and reduction in hospital mortality. Similarly, a comprehensive analysis of pooled data from the available literature in patients with sepsis and septic shock concluded that administration of antibiotics within three hours of ED triage or one hour of recognition of sepsis/septic shock did not confer a mortality benefit [5]. These results suggest that the currently recommended specific timing metrics in international guidelines are not supported by the currently available evidence [5].

In this study, an audit on the interval from recognition of sepsis or septic shock in inpatients to initial administration of antibiotics is analyzed. Consequently, the institutional compliance towards the Surviving Sepsis Campaign’s recommendation (certified by MOH) for early antibiotic therapy is determined. The adherence is evaluated from the time the antibiotic is served right from endorsement to administration. A data collection form containing a direct yes or no scale is employed in this study to assess the association between the antibiotic administration timing and mortality rate.

## 2. Results

Throughout the six-month study period, a total of 525 patients were admitted into ICU and approximately 78 of them were recruited in this study, comprising 46 (59%) males and 32 (41%) females. All the recruited patients fulfilled the study inclusion criteria. The demographic and clinical characteristics of the patients are presented in Table 1, whereas Table 2 and Table 3 depict the further analysis.

### 2.1. Audit on the Time Interval

Of all the 78 patients, only 38 (48.7%) of them received the antibiotics prescribed within an hour while the remaining 40 (51.3%) patients were administered after an hour. Subsequent analysis on the potential causes of not administrating the antibiotics within an hour included waiting for culture to be taken (46.8%, *n* = 22), delay in sending the order (40.4%, *n* = 19), not prescribed with a stat dose (10.6%, *n* = 5) and a patient had an ongoing procedure (2.1%, *n* = 1). Nursing staff accounted for 47.8% (*n* = 22) of potential sources of not delivering the antibiotics within an hour, followed by supporting staff (39.1%, *n* = 18), and physicians (13%, *n* = 6).

### 2.2. Compliance with the Surviving Sepsis Campaign

The compliance rate as per the Surviving Sepsis Campaign was only 51.3%, which accounts for 40 patients. Another 48.7% of them failed to comply with the campaign.

### 2.3. Association between Antibiotic Administration Timings and Mortality Rate

The reported mortality rate after 48 h was (39.7%, *n* = 31) and the majority of the study participants survived (60.3%, *n* = 47). Further analysis revealed that delayed antibiotics administration (more than an hour) was significantly associated with an increase in the mortality rate.

## 3. Discussion

This study analyzed the interval from recognition of sepsis or septic shock in inpatients to the initial administration of antibiotics. Early administration of antibiotics was significantly associated with lower mortality after 48 h. This finding coincides with the outcome reported in other studies conducted elsewhere, thereby supporting the clinical significance of prompt administration of antibiotics as recommended in the Surviving Sepsis Campaign.

A study by Kumar et al. found an average 7.6% decrease in survival with every hour delay in receiving antibiotics after the onset of hypotension. This study was conducted retrospectively involving patients with sepsis admitted to ICUs from 1989 to 2004 [6]. Vincent et al. also found that the median time to antibiotic administration was 2.1 h (interquartile range, 1.4–3.1 h). The adjusted odds ratio for hospital mortality based on each hour of delay in antibiotics after registration was 1.09 (95% confidence interval (CI), 1.05–1.13) for each elapsed hour between registration and antibiotic administration. The increase in absolute mortality associated with an hour delay in antibiotic administration was 0.3% for sepsis, 0.4% for severe sepsis, and 1.8% for shock [7]. A similar study conducted by Ferrer et al. involving patients with severe sepsis and septic shock in 165 ICUs concluded that the time of administration was positively associated with the mortality rate [8].

No doubt there are also controversies regarding the timing of antibiotics. Only one study conducted in Japan showed that the overall crude mortality rate was 23.4%, where patients in the 0–60 min group had the highest mortality (28.0%) while those in the 61–120 min group had the lowest mortality (20.2%). The researchers further concluded that there is no association between earlier antibiotic administration and reduction in hospital mortality in patients with severe sepsis [9].

Various studies have demonstrated that early administration of antibiotics is the key component to improving the survival rate. Previous serial Surviving Sepsis Campaign guidelines have repeatedly recommended early administration of antibiotics, preferably within 1 h upon sepsis diagnosis [2]. A randomized control trial (MEDUSA study) revealed that the mortality rate possibly increases by 2% in every hour delay of antibiotics administration [10]. Hence, administrating prescribed antibiotics within the recommended time frame should be regarded as one of the emergency treatments.

It is obvious that multiple reasons are listed which makes timely administration of the antibiotics even more challenging. In this study, only 48.7% (38 patients) of the patients had their antibiotics served within an hour. Meanwhile, more than 50% of the cases were given antibiotics an hour later. Other confounding factors, such as difficult blood taking, manpower, experience levels of the staff, and lack of awareness among the healthcare workers were not considered in this study. These events may cause a delay in blood collection, thereby contributing to most delays in administrating antibiotics. Necessary measures were taken to tackle the problems identified. For instance, a medical house officer was specifically assigned to take cultures to further facilitate the early administration of antibiotics. Awareness of the latest sepsis guidelines was stressed among all healthcare workers (doctors, nurses, and supporting staff) in our hospital by giving continuous medical education (CME) throughout the year and during ward rounds to explain the benefit of timely antibiotic administration. As for the delay in prescribing by physicians, no further investigation was undertaken to identify the cause of the delay. Other few possible causes include antibiotics being ordered by the physicians but the antibiotic order form was not attached at the time of order; physicians being unfamiliar with some of the less commonly used antibiotics which may delay the timing of prescription; and there may be patient complexity which makes the diagnostic and choice of antibiotics more challenging. Significant antibiotic prescriptions (39%) were in usual order rather than stat and usual order. The prescriptions without STAT (immediate) word may contribute significantly to the delay in antibiotics administrations as supporting staff would just administer the antibiotics according to the usual timely manner. Thus, re-education on the importance of appropriate prescriptions needs to be carried out. In order to overcome this issue, a digital system has been recently introduced, whereby physicians just need to prescribe via the online system. This online ordering platform eases the physicians’ task by being just one click away. Of all possible factors listed above, self-awareness among team members is suggested to play a crucial role in the early administration of antibiotics. Early administration of antibiotics upon diagnosis of sepsis needs to be included as part of the health care key performance indicators (KPI). With reference to the results after endorsing early thrombolytic therapy for myocardial infarction as part of the KPI for health care facilities, such a measure will assist in resolving the problems of delaying antibiotics administration. At the same time, roadshows of early appropriate antibiotic administrations need to be performed regularly to increase self-awareness and knowledge.

On another separate note, a prospective study (i.e., point prevalence survey (PSS)) was conducted by the Ministry of Health, Malaysia at all the facilities including the present study center in 2019. The survey focused on several factors including the appropriateness of the choice of antibiotics for causative pathogens in various types of infections. The survey was created against the National Antibiotic Guideline (NAG) 2019 developed by the ministry itself. Resultantly, the ‘appropriateness’ in terms of the choice of antibiotic for various infections was 82%, which is fairly acceptable and provides an additional benefit in this study. This revealed that the choices of antibiotics were not a distinctive issue but rather the timely antibiotic administration remains an unresolved issue in this facility.

Being a single-center study, there are certainly possibilities of biases and insufficient data to establish significant direct cause and effect relationships. Hence, the results obtained in this study may not be appropriate to generalize to other centers given various patient populations, staffing strengths, and limited antibiotics distribution. Furthermore, the sample size in this study was considerably small. Perhaps, a larger sample size involving multiple study centers is required to further validate the results obtained from this study. Including patients presenting to the emergency department with sepsis would have strengthened this research. To improve the whole ICU sepsis survival rate, we believe this will be a teamwork revolution rather than an ICU team alone. Based on current clinical practice, we believe there is a significant difference in the interval to initiation of antibiotics among patients with onset of sepsis in the wards, ICU, and emergency department.

## 4. Materials and Methods

### 4.1. Study Design and Setting

This study was conducted prospectively at a single center in an intensive care unit (ICU) facility with 17 beds located in Penang, Malaysia. This tertiary-care center has a 393-bed capacity, which serves a population of approximately 900,000. Ethical board approval by MREC was obtained prior to the initiation of this study.

### 4.2. Data Definitions

Sepsis was defined as suspected or confirmed infection in the presence of two or more systemic inflammatory response syndrome criteria. The systemic inflammatory response syndrome was defined by two or more of the following conditions: (1) body temperature greater than 38 °C or less than 36 °C; (2) heart rate greater than 90 beats per minute; (3) respiratory rate greater than 20 breaths per minute or PaCO_2_ of less than 32 mmHg; and (4) white cell counts greater than 12,000/mm^3^, less than 4000/mm^3^, or the presence of more than 100% immature neutrophils (‘bands’) [11]. Meanwhile, septic shock was defined as sepsis that presented with hypotension (systolic blood pressure < 90 mmHg, mean arterial pressure (MAP) < 60 mmHg, or a reduction in systolic blood pressure of >40 mmHg from baseline) despite adequate fluid resuscitation, in the absence of other causes for hypotension with acute organ dysfunction. Once sepsis or septic shock had been recognized, the clinicians would have to initiate a broad-spectrum antibiotic (e.g., Carbapenem). The 2017 Surviving Sepsis Campaign guidelines endorse a focus on the rapidity of treatment once sepsis has been identified, with a strong recommendation for the administration of antimicrobial drugs within 1 h. Hence, the time frame meant in this study is from the time sepsis recognition to the initiation of antibiotics.

### 4.3. Study Population

Patients were identified daily from routine rounds by anesthetists for six months from March 2020 to August 2020. All the patients, regardless of their age, gender, and admission date, who were admitted into ICU and had their first episode of sepsis or septic shock concomitantly were placed on a broad-spectrum antibiotic given intravenously. Data from the very first episode of sepsis in patients who had multiple episodes of sepsis throughout the study period were included in this study. Two medical officers were assigned to facilitate the data collection process by completing the data collection sheet on a daily basis. On the other hand, patients who had received antibiotics even before the development of sepsis or shock and also patients with undocumented data on antibiotics use were excluded from this study. The calculated sample size was *n* = 73 with a confidence interval of 95% and 5% random margin of error.

### 4.4. Data Collection

A separate datasheet was used in Microsoft Excel format. The data collected included patients’ demographic characteristics, underlying comorbidities, mortality after 48 h, medication administration timings, potential causes, SOFA scores every 48 h and possible sources for antibiotics not being served within an hour.

### 4.5. Statistical Analysis

Statistical analysis was performed using SPSS version 22. Descriptive analysis was used to summarize the collected data into median (interquartile ranges (IQR)), mean ± standard deviation (SD), or frequencies (%) as appropriate. Multiple logistic regression was used to compare dichotomous variables. Univariate analysis was performed to evaluate the mortality after 48 h. A *p*-value < 0.05 was considered statistically significant. All the patients enrolled were included in the primary analysis.

### 4.6. Outcome Measurement

This study was performed to audit the interval from recognition of sepsis or septic shock in warded patients to the initial administration of antibiotics. This would further assess the institutional compliance with the Surviving Sepsis Campaign’s recommendation for early antibiotic therapy. Consequently, the association between antibiotic administration timings and mortality rates was determined.

## 5. Conclusions

This study revealed a significant difference in conferring the mortality benefit, if the administration of a broad-spectrum antibiotic exceeds an hour right from recognition of the onset of sepsis or septic shock to the time the antibiotic is administered, concomitantly parallel with the Surviving Sepsis Campaign.

## Figures and Tables

**Table 1 antibiotics-11-01202-t001:** Demographic and clinical characteristics of study patients.

Characteristics	No. (%) of Patients
**Age**	
<55 years	39 (50.0)
≥55 years	39 (50.0)
**Gender**	
Male	46 (59.0)
Female	32 (41.0)
**Ethnicity**	
Malay	48 (61.5)
Chinese	18 (23.1)
Indian	12 (15.4)
**With underlying Chronic Illness**	
No	35 (44.9)
Yes	43 (55.1)
**Mortality after 48 h**	
No	47 (60.3)
Yes	31 (39.7)
**Medication administered within 1 h from order time**	
No	40 (51.3)
Yes	38 (48.7)
**Potential cause of not delivering drug within 1 h**	
Awaiting culture	22 (46.8)
Delay in sending the order	19 (40.4)
No stat (immediate) dose is given	5 (10.6)
Procedure ongoing	1 (2.1)
**The potential source of not delivering drug within 1 h**	
Nursing	22 (47.8)
Supporting staff	18 (39.1)
Physician	6 (13.0)
**Diagnosis**	
Occult sepsis	44 (56.4)
Known sepsis	34 (43.6)
**Order status**	
Usual order	31 (39.7)
Stat and usual order	47 (60.3)
**After-office-hours order**	
No	49 (62.8)
Yes	29 (37.2)
**Source of drug stock**	
Pharmacy	58 (74.4)
ICU	20 (25.6)
**Administered drug category**	
Piperacillin–Tazobactam	39 (50.0)
Meropenem	26 (33.3)
Others	13 (16.7)
**Time taken for mediation delivery, median (IQR)**	1 h 42 min (3 h 17 min)

**Table 2 antibiotics-11-01202-t002:** Univariate analysis performed to evaluate mortality after 48 h.

Variables	Survived *N* (%)	Died *N* (%)	*p*-Value
**Age category**			0.817
<55 years	23 (59.0%)	16 (41.0%)	
≥55 years	24 (61.5%)	15 (38.5%)	
**Gender**			0.044 *
Male	32 (69.6)	14 (30.4)	
Female	15 (46.9)	17 (53.1)	
**Ethnicity**			0.064
Malay	24 (50.0)	24 (50.0)	
Chinese	14 (77.8)	4 (22.2)	
Indian	9 (75.0)	3 (25.0)	
**With underlying Chronic Illness**			0.374
No	23 (65.7)	12 (34.3)	
Yes	24 (55.8)	19 (44.2)	
**Medication administered within 1 h from order time**			0.018 *
No	19 (47.5)	21 (52.5)	
Yes	28 (73.7)	10 (26.3)	
**After-office-hours order**			0.480
No	31 (63.3)	18 (36.7)	
Yes	16 (55.2)	13 (44.8)	
**Diagnosis**			0.820
Occult sepsis	27 (61.4)	17 (38.6)	
Known sepsis	20 (58.8)	14 (41.2)	
**Type of infections**			0.542
CLABSI	24 (68.6)	11 (31.4)	
CAUTI	3 (100.0)	0 (0.0)	
SSI	4 (66.7)	2 (33.3)	
VAP	16 (47.1)	18 (52.9)	
**Order status**			<0.001 *
Usual order	27 (87.1)	4 (12.9)	
Stat and usual order	20 (42.6)	27 (57.4)	
**Source of drug stock**			0.302
Pharmacy	33 (56.9)	25 (43.1)	
ICU	14 (70.0)	6 (30.0)	
**SOFA Score**			0.202
Score 0 to 6	9 (100.0)	0 (0.0)	
Score 7 to 9	12 (100.0)	0 (0.0)	
Score 10 to 12	25 (86.2)	4 (13.8)	
Score 13 to 14	1 (14.3)	6 (85.7)	
Score 15	0 (0.0)	17 (100.0)	
Score 16 to 24	0 (0.0)	4 (100.0)	
**Administered drug category**			0.706
Tazosin	25 (64.1)	39 (50.0)	
Meropenem	14 (53.8)	26 (33.3)	
Others	8 (61.5)	13 (16.7)	
**Inotropic support**			0.031 *
No	2 (100.0)	0 (0.0)	
Yes	45 (59.2)	31 (40.8)	
**Artificial ventilation**			0.240
HFMO_2_	1 (100.0)	0 (0.0)	
CPAP	13 (76.5)	4 (23.5)	
SIMV	28 (77.8)	8 (22.2)	
BILEVEL	5 (20.8)	19 (79.2)	

*: Self explanatory.

**Table 3 antibiotics-11-01202-t003:** Multiple logistic regression on various variables.

Variables		Crude OR	(95% CI)	*p*-Value ^a^	Adj. OR	(95% CI)	*p*-Value ^b^
Age category	<55 years	1.00 (ref.)		0.817	1.00 (ref.)		0.525
	≥55 years	1.13	(0.45, 2.76)		1.50	(0.43, 5.22)	
Gender	Male	1.00 (ref.)		0.046	1.00 (ref.)		0.187
	Female	0.39	(0.15, 0.98)		0.41	(0.11, 1.54)	
Ethnicity	Malay	1.00 (ref.)		0.072	1.00 (ref.)		0.244
	Chinese	3.00	(0.72, 12.46)		3.06	(0.47, 20.15)	
	Indian	0.86	(0.15, 4.76)		1.18	(0.13, 11.08)	
With underlying Chronic Illness	No	1.00 (ref.)		0.375	1.00 (ref.)		0.711
	Yes	0.66	(0.26, 1.66)		0.79	(0.17, 4.05)	
Medication administered within 1 h from order time	No	1.00 (ref.)		0.020	1.00 (ref.)		0.015
	Yes	3.10	(1.19, 8.02)		5.79	(1.41, 23.78)	
After-office-hours order	No	1.00 (ref.)		0.481	1.00 (ref.)		0.826
	Yes	0.72	(0.28, 1.82)		0.87	(0.24, 3.09)	
Diagnosis	No	1.00 (ref.)		0.820	1.00 (ref.)		0.808
	Yes	0.90	(0.36, 2.24)		1.17	(0.34, 4.02)	
Order status	Usual order	1.00 (ref.)		<0.001	1.00 (ref.)		0.001
	Stat and usual order	0.11	(0.03, 0.36)		0.08	(0.02, 0.35)	
Source of drug stock	Pharmacy	1.00 (ref.)		0.305	1.00 (ref.)		0.817
	ICU	1.77	(0.60, 5.25)		0.83	(0.17, 4.08)	
Administered drug category	Tazosin	1.00 (ref.)		0.707	1.00 (ref.)		0.619
	Meropenem	1.37	(0.35, 5.33)		0.62	(0.10, 4.03)	
	Others	0.90	(0.25, 3.27)		0.74	(0.13, 4.16)	

^a^: Non adjusted confidence interval (CI); ^b^: Adjusted confidence interval (CI) based on age, gender, and ethnicity.

## Data Availability

Required data may be provided upon request.

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
