# Peer review of "The Impact of Antibiotics Administration on Mortality for Time in Sepsis and Septic Shock Patients including Possible Reasons for Delayed Administration in Malaysia"

_antibiotics, 2022, doi:10.3390/antibiotics11091202_

Round 1
Reviewer 1 Report
The term “Severe Sepsis” appears in the manuscript, however, according to the Society of Critical Care Medicine (SCCM) and the European Society of Critical Care Medicine (ESICM) all cases of sepsis should be considered as a severe disease, so that the term "severe sepsis" should be abolished [Singer M, Deutschman CS, Seymour CW, Shankar-Hari M, Annane D, Bauer M, et al. The Third International Consensus Definitions for Sepsis and Septic Shock (Sepsis-3). JAMA. 2016;315(8):801-10].
The reference used to define sepsis and septic shock [Levy MM, Fink MP, Marshall JC, et al. 2001 SCCM/ ESICM/ ACCP/ ATS/ SIS International Sepsis Definitions Conference. Crit Care Med 2003;31:1250-6] is outdated. Since in 2016 the Third International Consensus Definitions for Sepsis and Septic Shock (Sepsis-3) took place, which established the definitions that are currently used. Because data collection took place from March 2020 to August 2020, the SEPSIS-3 definitions should have been used.
As indicated by the authors, the fact that this is a single-center study and the small number of patients included (n=73) compromises the ability to generalize the results. Additionally, there are other more influential studies on the topic that have already been published, so the results of the study under analysis do not add new results that could raise discussions on the topic.
Author Response
Dear reviewer,
Thanks for the feedback. I have provided point-by-point response for each comments given.
Many thanks.
The term “Severe Sepsis” appears in the manuscript, however, according to the Society of Critical Care Medicine (SCCM) and the European Society of Critical Care Medicine (ESICM) all cases of sepsis should be considered as a severe disease, so that the term "severe sepsis" should be abolished [Singer M, Deutschman CS, Seymour CW, Shankar-Hari M, Annane D, Bauer M, et al. The Third International Consensus Definitions for Sepsis and Septic Shock (Sepsis-3). JAMA. 2016;315(8):801-10].
Response: Amended accordingly. Used the latest consensus on the definitions.
The reference used to define sepsis and septic shock [Levy MM, Fink MP, Marshall JC, et al. 2001 SCCM/ ESICM/ ACCP/ ATS/ SIS International Sepsis Definitions Conference. Crit Care Med 2003;31:1250-6] is outdated. Since in 2016 the Third International Consensus Definitions for Sepsis and Septic Shock (Sepsis-3) took place, which established the definitions that are currently used. Because data collection took place from March 2020 to August 2020, the SEPSIS-3 definitions should have been used.
Response: Amended accordingly. Used the latest consensus on the definitions.
As indicated by the authors, the fact that this is a single-center study and the small number of patients included (n=73) compromises the ability to generalize the results. Additionally, there are other more influential studies on the topic that have already been published, so the results of the study under analysis do not add new results that could raise discussions on the topic.
Response: Yes, certainly agree with the reviewer that there are more influential studies available on the similar topic. However, these studies were not conducted in South East Asia region, precisely Malaysia as the norm of practice is not similar with other regions. There are still numerous factors which attribute to the ignorance in administering the antibiotic within an hour. Therefore, this study has provided a real time local data and its significances.
Reviewer 2 Report
In the paper “The Impact of Antibiotics Administration on Mortality for Time in Severe Sepsis and Septic Shock Patients” the authors conducted prospective study in a single center of Intensive Care Unit (ICU) for six month period. They have found that from 78 enrolled patients, only 38 (48.7%) received the antibiotics within an hour. The overall survival rate was 60.3%. The study revealed that delayed antibiotics administration (more than an hour) significantly affects mortality.
Comments:
1. What is % of mortality rate for the septic patients that received antibiotics within an hour? Correlation between % of mortality rate and time of antibiotic administration should be shown as well as decrease in survival with every hour delay.
2. In the Table 1, Mortality after 48h means regardless of time of antibiotic start?
3. It is stated (Results 2.3.) that “further analysis revealed that delayed antibiotics administration (more than an hour) was significantly associated with an increase in 72 the mortality rate”. Where the results from this analysis have been shown?
4. In the study the authors have analyzed possible reasons of delayed antibiotic administration in their hospital. That should be reflected and included in abstract and title as well.
5. All study limitations should be stated!
6. As this experience from Malaysia is very valuable, country name should be added to the title.
Author Response
Dear reviewer,
Thanks for your feedback. I have provided point-by-point response to the comments given. The English language has been proofread by certified professionals (enclosed the certificate).
Many thanks.
Comments:
- What is % of mortality rate for the septic patients that received antibiotics within an hour? Correlation between % of mortality rate and time of antibiotic administration should be shown as well as decrease in survival with every hour delay.
Response 1: The % of mortality rate for the septic patients that received antibiotics within an hour is 26.3% (n=10) as shown in Table 2. The said correlation vs survival with every hour delay was not performed in this study as it was not planned earlier.
- In the Table 1, Mortality after 48h means regardless of time of antibiotic start?
Response 2: 48h meant from the time the antibiotic was started.
- It is stated (Results 2.3.) that “further analysis revealed that delayed antibiotics administration (more than an hour) was significantly associated with an increase in 72 the mortality rate”. Where the results from this analysis have been shown?
Response 3: The original statement is as below “Further analysis revealed that delayed antibiotics administration (more than an hour) was significantly associated with an increase in the mortality rate”. The result is shown under Table 2 (Univariate analysis).
- In the study the authors have analyzed possible reasons of delayed antibiotic administration in their hospital. That should be reflected and included in abstract and title as well.
Response 4: Included in the title. Unable to include in the abstract due to limited word count.
- All study limitations should be stated!
Response 5: All the acknowledged study limitations by the authors are stated under discussion (last paragraph).
- As this experience from Malaysia is very valuable, country name should be added to the title.
Response 6: Amended accordingly.
Reviewer 3 Report
The study addresses a very interesting topic, the impact of early administration of antibiotics on mortality rates of patients with sepsis. However, there are some major concerns that should be addressed before considering the paper for publication.
1. The number of patients included is low and insufficient to support the authors’ conclusion on the impact of the timing of administration of antibiotics on mortality. Furthermore, also at least 7-day mortality should be assessed. Data should be rather presented as an audit on the compliance with SSC guidelines.
2. The study was conducted in 2020 but authors used the categories of severe sepsis and septic shock following 2001 SCCM/ ESICM/ ACCP/ ATS/ SIS definitions, that in 2020 have been already overcome by those of sepsis and septic shock following the Third International Consensus Definitions for Sepsis and Septic Shock (Sepsis-3) published in 2016; most recent definitions should be used.
3. Some key information which can have an impact on the mortality of patients with sepsis are missed (e.g. community acquired or nosocomial sepsis, the presence of comorbidities should be assessed using a score system -e.g. Charlson Comorbidities index-, no microbiological data are provided).
Author Response
Dear reviewer,
Thanks for your feedback. I have provided point-by-point response to the comments given.
Many thanks.
- The number of patients included is low and insufficient to support the authors’ conclusion on the impact of the timing of administration of antibiotics on mortality. Furthermore, also at least 7-day mortality should be assessed. Data should be rather presented as an audit on the compliance with SSC guidelines.
Response 1: Yes, certainly agree with the reviewer that unable to generalize the findings as the sample size is small. However, similar studies were not conducted in South East Asia region, precisely Malaysia as the norm of practice is not similar with the other regions. There are still numerous factors which attribute to the ignorance in administering the antibiotic within an hour. Therefore, this study has provided a real time local data and its significances. Shorter duration was chosen for mortality determination as 24-48 hours is crucial for survival especially when it involves infection. Otherwise, prognosis worsens beyond this time. Data was not presented as an audit as this study found to have significance association between administration timing and mortality with the presence of vasopressor support and etc.
- The study was conducted in 2020 but authors used the categories of severe sepsis and septic shock following 2001 SCCM/ ESICM/ ACCP/ ATS/ SIS definitions, that in 2020 have been already overcome by those of sepsis and septic shock following the Third International Consensus Definitions for Sepsis and Septic Shock (Sepsis-3) published in 2016; most recent definitions should be used.
Response 2: Amended accordingly. Used the latest consensus on the definitions.
- Some key information which can have an impact on the mortality of patients with sepsis are missed (e.g. community acquired or nosocomial sepsis, the presence of comorbidities should be assessed using a score system -e.g. Charlson Comorbidities index-, no microbiological data are provided).
Response 3: Microbiological data was not provided as it was given empirical and every subjects were studied until 48 hours only. The samples recruited in this study have almost the similar comorbidities and hence the impact on mortality rate remains almost similar (no outliers with different comorbidities). Instead of community or nosocomial sepsis, it was grouped as types of infections emphasizing on the origin of the infection.
Round 2
Reviewer 1 Report
None
Reviewer 2 Report
The authors answered my comments and concerns. I have no more questions for the authors.